# Vemurafenib and Rituximab in Patients with Hairy Cell Leukemia Previously Treated with Moxetumomab Pasudotox

**DOI:** 10.3390/jcm10132800

**Published:** 2021-06-25

**Authors:** Tadeusz Robak, Agnieszka Janus, Krzysztof Jamroziak, Enrico Tiacci, Robert J. Kreitman

**Affiliations:** 1Department of Hematology, Medical University of Lodz, 93-510 Lodz, Poland; 2Copernicus Memorial Hospital, 93-510 Lodz, Poland; agnieszka_janus@poczta.onet.pl; 3Department of Hematology, Transplantation and Internal Medicine, Medical University of Warsaw, 02-097 Warsaw, Poland; krzysztof.jamroziak@wp.pl; 4Institute of Hematology, University and Hospital of Perugia, 06132 Perugia, Italy; enrico.tiacci@unipg.it; 5Laboratory of Molecular Biology, Clinical Immunotherapy Section, National Cancer Institute, National Institutes of Health, 9000 Rockville Pike, Bethesda, MD 20892, USA; kreitmar@mail.nih.gov

**Keywords:** *BRAF*, cladribine, dabrafenib, hairy cell leukemia, moxetumomab pasudotox, rituximab, vemurafenib

## Abstract

The purine nucleoside analogues cladribine and pentostatin are highly-active first-line therapeutic treatments for hairy cell leukemia (HCL), resulting in complete response rates of 80% to 90%. However, HCL patients continue to relapse, and sooner or later, most require subsequent lines of treatment. This report presents the cases of four relapsed patients with classic HCL who were treated with vemurafenib (mostly at the low dose of 240 mg twice daily for 16 weeks) combined with rituximab after the failure of several lines of therapy including cladribine with or without rituximab and moxetumomab pasudotox. Two patients achieved minimal residual disease negative complete response after combined treatment with vemurafenib and rituximab, with a hematologic response ongoing after 38 months from the end of treatment in one patient and a relapse of cytopenias occurring after 13 months in the other patient. A third patient normalized her blood counts and this hematologic response, which was not evaluated in the bone marrow at the end of treatment, was lost after 18 months. The last patient died due to infection and multi-organ failure, too early to verify response to vemurafenib. Two patients who had relapsed after vemurafenib and rituximab derived meaningful clinical benefit from retreatment with the same agents, but eventually relapsed again and started indefinite therapy with dabrafenib and trametinib leading to normalization of the blood counts (despite heavy bone marrow infiltration in the only patient so far evaluable in that regard). The outcomes of these cases indicate that novel targeted agents and, in particular, vemurafenib, combined with rituximab, improve the prognosis of HCL patients, even those heavily pretreated with PNAs and moxetumomab pasudotox.

## 1. Introduction

Hairy cell leukemia (HCL) is a rare chronic B-cell neoplasm originating from a mature B lymphocyte; it is characterized by marked splenomegaly, progressive pancytopenia, and infiltrations of the bone marrow (BM) with reactive marrow fibrosis [1,2]. The annual incidence of HCL is estimated to be 0.3 cases per 100,000 and the disease constitutes 2–3% of all leukemias [3,4]. A diagnosis of HCL is established by a combination of peripheral blood smear, flow cytometry and bone marrow biopsy. Recently, it has been found that the *BRAF* p.V600E somatic mutation is present in 90% to 100% of patients with classic HCL, and this has been described as a disease-defining genetic event [5]. The drugs of choice in the treatment of HCL are purine nucleoside analogs (PNAs), cladribine (2-CdA) and pentostatin (DCF) [1,2]. These agents induce durable complete response (CR) in more than 70% of first line patients and are also effective in re-induction therapy. However, patients continue to relapse, with median response duration of 8–10 years, and most experience several relapses [6,7]. Repeated treatments with the same PNA are effective in some patients, but the CR rate and response duration following subsequent courses typically shorten with subsequent treatments [7,8,9]. Patients who relapse within two or three years after the first PNA treatment have worse prognosis and lower likelihood of a second durable CR achievement. In addition, in some cases, the disease is refractory to PNA therapy. The median duration of response to second line 2-CdA monotherapy is 2–9 years [8,9].

Moxetumomab pasudotox (Moxe, LUMOXITI™, AstraZeneca, Wilmington, DE, USA) is a recombinant immunotoxin composed of the Fv fragment of an anti-CD22 monoclonal antibody fused to a 38-kDa fragment of *Pseudomonas* exotoxin (PE38) [10]. Its approval was based on phase 1 and phase 3 clinical trials evaluating the safety and efficacy of the drug in relapsed/refractory HCL [10,11,12]. The *BRAF* V600E mutation inhibitors vemurafenib and dabrafenib also exhibit remarkable activity in multiply-relapsed and refractory patients with classic HCL [13,14,15]. Combining *BRAF* inhibitors with anti-CD20 monoclonal antibodies or MEK inhibitor as therapy can be even more effective than using the *BRAF* inhibitors alone [16,17].

However, the optimal drug sequencing for relapsed/refractory HCL remains unestablished. The present study reviews data from four heavily-pretreated HCL patients who received combination therapy of vemurafenib and rituximab after the failure of Moxe.

## 2. Materials and Methods

This retrospective study examines the cases of four patients with classic HCL who were treated with vemurafenib and rituximab following relapse after Moxe. Their clinical records were identified in the hospital registry at Copernicus Memorial Hospital (Lodz, Poland). All patients were found to be positive for *BRAF* p.V600E, as indicated by conventional AS PCR for the *BRAF* V600E mutation; this has an analytical sensitivity 0.1% mutated allele [18]. AE were evaluated according to Common Terminology Criteria for Adverse Events (CTCAE) Version 5.0. Common Terminology Criteria for Adverse Events (CTCAE), Version 5 Ed, National Cancer Institute, 2017 (http://evs.nci.nih.gov/ftp1/CTCAE/About.html; accessed on 21 June 2021) Response to therapy was evaluated using the consensus guidelines for the diagnosis and management of patients with HCL [1]. All procedures performed in studies involving human participants were in accordance with the ethical standards of the institutional and/or national research committee and with the 1964 Helsinki Declaration and its later amendments or comparable ethical standards.

## 3. Results

The demographics, prior treatments and clinical outcomes of the four patients are summarized in Table 1.

Before Moxe initiation, all patients had been previously treated with at least three lines and one patient with six lines of therapy. Previous treatments included 2-CdA monotherapy, 2-CdA combined with rituximab, and IFN-α. The median time from diagnosis to Moxe initiation was 112 (range 28–258) months. All four patients were treated with Moxe at 40 micrograms on days 1, 3, and 5 of each 28-day cycle for up to five to six courses within a phase 3 clinical trial. Two patients achieved a CR and two PR, with hematologic remission lasting 11 to 38 months from the end of treatment (Table 1). None of the four patients experienced serious adverse reactions, such as capillary leak syndrome or hemolytic uremic syndrome. In all four patients, vemurafenib plus rituximab was given as the subsequent treatment after Moxe due to HCL progression, with no serious adverse reactions from this combo regimen. One patient (No 4), who started vemurafenib at the dose of 960 mg BID, was not evaluable as he died early of a severe infection (after having survived for 302 months from HCL diagnosis). In the other three patients, the drug was given at the low dose of 240 mg BID for 16 weeks in combination with rituximab (375 mg/m^2^/d/iv) every two weeks for eight doses. Two patients achieved MRD negative CR with hematologic remission that was ongoing at 38 months from the end of treatment in one case, while it was lost after 13 months in the other case. The third patient achieved a hematologic remission (with unknown bone marrow response at the end of treatment), that was lost after 18 months. The two patients who progressed after vemurafenib plus rituximab were successfully retreated with this regimen and derived clinical benefit for a duration that was similar (in one patient) and lower (in the other patient) compared to the previous course. At eventual relapse, both patients were started on indefinite dabrafenib and trametinib; one of these two patients (No. 3) regained a hematologic remission since 10 months despite persistence of marked leukemic infiltration of the bone marrow, while the other patient (No. 2) recently started treatment and achieved hematologic remission that was ongoing at 2 months.

### 3.1. Case Description

#### 3.1.1. Case 1

A 28-year-old woman with pancytopenia, symptoms of hemorrhagic diathesis and splenomegaly (160 mm) was first seen in July 2008. A diagnosis of HCL was confirmed by immunohistochemistry and flow cytometry. Between August 2008 and April 2010 she received two lines of 2-CdA based regimens (Table 1). She relapsed again in April 2010 and second-line treatment was administered, consisting of 2-CdA and rituximab and a PR was achieved. The second relapse was noted in February 2011 and she started interferon alpha (IFN-α) at a dose of 3 mln IU, 3× per week given for five months, and obtained only a PR lasting until June 2012. In September 2012, fourth-line treatment with 2-CdA was applied, and again a PR was achieved. In April 2015, the fifth-line treatment was begun, with six cycles of Moxe within the phase 3 clinical trial at the National Institutes of Health (NIH) Clinical Center [11,12]. At the end of the treatment, she fulfilled the criteria for PR. In August 2017, she experienced the fifth disease relapse, with hemoglobin (Hb) 12.3 g/dl, white blood cells (WBC) 2.32 × 10^3^/µL, absolute neutrophil count (ANC) 0.83 × 10^3^/µL and platelets (PLT) 73 × 10^3^/µL. The presence of *BRAF* p.V600E mutation was confirmed by AS-PCR in the PB and BM aspirate and she started treatment with vemurafenib (240 mg BID for 4 months) in combination with rituximab (375 mg/m^2^/d/iv) every two weeks for eight doses. During the treatment, she complained of diffuse arthralgia and skin rash. In response to these symptoms, concomitant treatment with prednisone 10 mg/d/orally was initiated, with success. CBC recovery was observed after one month of treatment. After the end of the therapy in January 2018 she obtained a MRD negative complete response (CR) confirmed by an assessment of the BM aspirate and flow cytometry, BM trephine biopsy and AS-PCR for *BRAF* p.V600E mutation in the peripheral blood (PB) and BM aspirate (with a sensitivity threshold of 0.1% mutant alleles). As of April 2021, 38 months after treatment discontinuation, the patient has remained without any evidence of the disease with normal CBC.

#### 3.1.2. Case 2

A 33-year-old woman was diagnosed with symptomatic HCL in June 2013. Between June and January 2014, she received two courses of 2-CdA, which resulted in a PR (Table 1). Subsequently, the patient relapsed, and second-line treatment with IFN-α 3 mln U × 3 per week was administered from September 2014 to December 2014 without response. The disease progressed in January 2015, and she began third-line treatment with 2-CdA in combination with eight doses of rituximab and a PR was obtained, which lasted only six months. In September 2015, she started Moxe for six courses within a pivotal multicenter trial and again PR was achieved. In November 2016 she experienced the fourth relapse, with progressive pancytopenia including transfusion-dependent anemia and infections. In January 2017 she started the fifth-line treatment with vemurafenib 960 mg BID and rituximab 375 mg/m^2^/d/iv scheduled every two weeks for eight doses. After the initial two weeks, the dose of vemurafenib was reduced to 240 mg BID due to intolerance (photosensitivity, arthralgia, musculoskeletal pain, rash); treatment was continued for a further 16 weeks without any further complications. Cytopenias resolved after one month from the start of treatment. In January 2019, she was admitted to the emergency room because of fever and fatigue. Her CBC revealed WBC 0.91 × 10^3^/µL, ANC 0.54 × 10^3^/µL, Hb 8.1 g/dL and PLT 135 × 10^3^/µL. Bone marrow aspirate and trephine biopsy confirmed the relapse of HCL. In February 2019, retreatment with 240 mg BID vemurafenib for 16 weeks with eight doses of 375 mg/m^2^/d/iv rituximab (one dose every two weeks) was initiated. To prevent previous noted vemurafenib toxicities, 10 mg prednisone was initiated. PB counts normalized four weeks after retreatment initiation, and a CR MRD (+) was documented in a bone marrow evaluation done in October 2019 about 4 months after the end of treatment, with the *BRAF* p.V600E mutation present at an allele frequency of 0.085% by digital PCR. She remained in remission for 17 months. In February 2020, progression of HCL was diagnosed with pancytopenia and 80% of hairy cell involvement in BM trephine biopsy. In April 2021 she started combined therapy with dabrafenib (Tafinlar) 75–150 mg/d BID p.o. and trametinib (Mekinist) 2 mg/day. Currently she has been maintained on these drugs for two months, achieving a hematological response.

#### 3.1.3. Case 3

A 53-year-old man was first seen in our Department in July 2003 because of fever and pancytopenia. Bone marrow biopsy confirmed a diagnosis of classic HCL by immunohistochemistry and flow cytometry. Between July 2003 and April 2007 the patient achieved three courses of 2-CdA and a CR was obtained after each course (Table 1). Progression after a third course of 2-CdA was reported in 2015 and he started Moxe as a part of a phase 3 clinical trial and the patient obtained a MRD negative CR. In October 2017, he was admitted to our hospital because of anemia, thrombocytopenia, and infection. BM aspirate was consistent with 60% infiltration with HCL cells, and the presence of *BRAF* V600E mutation was confirmed by PCR. Between November 2017 and February 2018, the patient was treated with vemurafenib (240 mg BID) and 375 mg/m^2^/d/iv rituximab every 2 weeks for eight doses. Blood counts normalized within 1 month. MRD-negative CR (at a sensitivity threshold of 0.1% mutant *BRAF*-V600E alleles) was obtained after the end of treatment. However, the patient relapsed in February 2019, with massive splenomegaly (320 × 120 mm) and pancytopenia. In March 2019 retreatment was started with vemurafenib at the dose 240 mg BID for 7 weeks in combination with rituximab 375 mg/m2/iv for a total of 4 doses administered every two weeks. After one month blood counts improved without, however, any reduction of splenomegaly. The patient was splenectomized in May 2019. As a result of this salvage therapy, short-termed hematological remission was obtained. In September 2019 the disease recurred with pancytopenia and gastrointestinal tract bleeding as a result of severe thrombocytopenia. In October 2019, retreatment with vemurafenib (240 mg BID) and rituximab (eight doses, one every other week) was started again and continued for 16 weeks until February 2020, without any toxicities. The CBC returned to normal at the end of the treatment. He relapsed again in June 2020 and combined treatment with dabrafenib (150 mg BID po) and trametinib (2 mg/day) was initiated. Two weeks later, the dose of both drugs was reduced (dabrafenib 75 mg BID, and trametinib 2 mg every other day) due to progressive pancytopenia. Hematologic remission was first achieved 2 months after starting treatment and was maintained until April 2021, when he developed mild thrombocytopenia and trephine biopsy showed 80% leukemic infiltration.

#### 3.1.4. Case 4

A 45-year-old man was first seen with pancytopenia and splenomegaly in June 1994. A bone marrow (BM) biopsy was consistent with diagnosis of classic HCL by immunohistochemistry and flow cytometry. Between October 1994 and May 2013 the patient was treated with five courses of 0.12 mg/kg/d/iv 2-CdA for five days each, resulting in a PR, PR, CR CR and PR, respectively (Table 1). After the fourth relapse, in January 2016, the patient was included in the clinical trial with Moxe. Following this sixth course of treatment, BM biopsy revealed CR MRD negative. In August 2019, he was admitted to our Department with progressive pancytopenia. BM biopsy confirmed progression of HCL with 90% marrow involvement, and the *BRAF* p.V600E mutation was detected. Salvage treatment was initiated with vemurafenib 960 mg orally twice a day and one dose of rituximab (375 mg/m^2^/d/iv) simultaneously with filgrastim. Considering the profound decline in ANC, rituximab was withheld. However, three weeks later, the patient developed pneumonia and septic shock with blood cultures positive for multi-resistant Pseudomonas aeruginosa and died of multi-organ failure.

## 4. Discussion

Conventional treatment with PNAs has been shown to provide a long-lasting hematological and clinical response in most patients with classic HCL. However, the patients often relapse and require multiple treatments, and may thus become refractory to re-treatment with PNAs. Fortunately, the recent introduction of novel agents has expanded the spectrum of therapy possibilities for those patients, and now, treatment options for the management of early-relapsed patients, or those refractory to PNAs, inlude IFN-α, rituximab and Moxe [19,20,21,22]. We reviewed the records from four patients treated at Copernicus Memorial Hospital, Lodz, Poland; all received multiple lines of 2-CdA and were subsequently treated with Moxe, followed by vemurafenib combined with rituximab. In addition, two patients were treated with IFN-α after relapse with 2-CdA: one of them did not respond and one obtained only a PR lasting 17 months. IFN-α may still have a place in the treatment, but its use is currently limited, being restricted to pregnant patients and in patients presenting with neutropenia below 0.2 × 109/L, when the risk of infection is high [20,21]. Our patients received Moxe within the pivotal, phase 3 multicenter study [11,12]. All patients responded to the treatment, two with PR and two with CR, with hematologic remission lasting from 11 to 38 months after the end of treatment. These results agree with those of a previous phase 1 and phase 3 trial, which included similar patient populations (with a median of 3 prior therapies): these trials demonstrated a durable CR of 30% to 64%, and an objective response rate (ORR) of 75% to 88% of the patients [10,22]. In addition, 82% to 85% of complete responders achieved MRD negativity [12,22]. Moxe has been approved by FDA for the treatment of patients with relapsed or refractory HCL after previous treatment with at least two lines of therapy, including at least one PNAs [23]. Moxe is now being tested in combination with rituximab to improve/hasten response and prevent immunogenicity.

More recently, vemurafenib, dabrafenib, trametinib and ibrutinib are currently under investigation in patients unsuitable for PNAs [13,24,25,26,27,28,29]. These agents have been proposed as relatively safe drugs, even in pretreated patients, with severe neutropenia and (in the case of *BRAF* inhibitors) infections. However, they are still considered “off-label” in the treatment of this disease.

In particular, the *BRAF* inhibitor vemurafenib was evaluated in relapsed and refractory patients with classic HCL in two phase 2 multicenter studies [13]. The drug was administered at a dose of 960 mg, twice daily for a median of 16–18 weeks. The OR rates were 96% to 100% with CR rates of 35% to 42%, and responses were obtained after a median of 8 to 12 weeks. Interestingly, while vemurafenib had been originally suggested to have similar effectiveness when used at much lower doses in a retrospective case series [14], an update of the same series expanded with additional patients showed that doses of vemurafenib lower than 480 mg bid might have suboptimal efficacy [15]. In particular, patients who received high-dose *BRAFi* treatment (vemurafenib ≥ 480 mg bid, dabrafenib ≥ 150 mg bid) demonstrated significantly longer treatment-free survival (median, 14.6 months) than patients who were treated with lower doses (median, 9.4 months). In a recently published phase 2 trial on 30 HCL patients with a median of three prior therapies (not including Moxe; [17]), adding rituximab (8 doses, one every other week) concomitantly and subsequently to a short treatment with vemurafenib at the dose of 960 mg bid for a total of 8 weeks produced a very high rate of complete remissions (87%) and negativity for minimal residual disease/MRD (60%). These deep responses translated into a durable relapse-free survival of 85% at almost 3 years of follow-up from the end of treatment. Relative dose intensity of vemurafenib was high (median of 92%), attesting to the safety and tolerability of the 960 mg bid dose when given for a short period (8 weeks in total). Receipt of a relative dose intensity <60% seemed to be associated with treatment failure, with the caveat of the low number of these events [17].

This is the first report documenting the activity of vemurafenib plus rituximab even in the context of relapse after Moxe, including hematologic remission in all three evaluable patients and MRD-negative CR in the two patients who underwent bone marrow evaluation at the end of treatment. Relapse-free survival after the end of treatment was substantial in one case (response ongoing at 38 months, versus 22 months after previous Moxe), while it was lower but still clinically meaningful in the other two cases (18 and 13 months, comparing well with 11 and 17 months after previous Moxe, respectively). When contrasted to the clinical trial of vemurafenib plus rituximab [17], the relapse-free survival in 2 of these 3 patients may appear somewhat lower, which might be due to a higher intrinsic drug resistance of their disease relapsing even after Moxe and/or (as suggested by [15]) to a lower efficacy of the vemurafenib dose used (240 mg bid). In patient 3, who relapsed 13 months after obtaining a CR with no MRD detected at a threshold of 0.1% *BRAF*-V600E mutant alleles on conventional PCR, it is also possible that MRD was actually present below that threshold and that a more sensitive approach such as digital PCR could have identified it.

The fourth patient, who had received 7 lines of therapy over 15 years, was not evaluable as he died too early (3 weeks after starting vemurafenib plus rituximab) of a severe infection caused by a pre-existing deep neutropenia, which even such a high effective treatment was not expected to necessarily resolve earlier (indeed, the median time to resolution of neutropenia after vemurafenib plus rituximab is 4 weeks [17]). Nevertheless, previous reports based on small numbers of cases suggested that vemurafenib is a safe and effective drug even in patients with HCL presenting with severe neutropenia, fever and infection [26,27,30].

This report also starts to address the issue of relapse after vemurafenib plus rituximab. Re-treatment with vemurafenib is an option that should be considered in this context, as exemplified by two of our cases enjoying clinical benefit from retreatment, although for a shorter duration in one case. Likewise, Liebers et al. reported rapid hematologic improvement in 17 patients retreated with *BRAF* inhibitors (85% previously treated with vemurafenib or dabrafenib) with at least one further *BRAFi* course [15]; however, the subsequent *BRAF* inhibitor courses yielded shorter response durations than the initial treatment. Similar results were obtained with vemurafenib retreatment in clinical trials [13]. Dabrafenib is another *BRAF* inhibitor that can be used in patients with heavily-pretreated recurrent/refractory *BRAF* V600E-mutated HCL, including cases previously treated with vemurafenib [24]. Dabrafenib can be also combined with the mitogen-activated extracellular signal-regulated kinase (MEK) inhibitor trametinib. Kreitman et al. found combination therapy of indefinite duration (median of 17 months) based on dabrafenib (150 mg twice daily) with the MEK inhibitor trametinib (2 mg once daily) is effective and well tolerated in 43 relapsed/refractory HCL patients with presence of a *BRAF* V600E mutation [17] who were not previously treated with a *BRAF* inhbitor. The OR rate was 78% including 49% CRs; Grade 3/4 AEs were observed in 49% of patients. Interestingly, our two patients who relapsed after retreatment with vemurafenib plus rituxmab both regained a hematologic remission with dabrafenib plus trametinib.

We used vemurafenib for limited time as in the most previous studies of this drug [13,14,17]. Limited treatment duration can decrease adverse events noted with *BRAF* inhibitors including skeletal pain, photosensitivity, skin tumors and renal toxicity. On the other hand, treatment with vemurafenib, especially in combination with rituximab induces deep, long responses making unlimited treatment redundant [17]. In addition, relapsed patients can achieve response after retreatment with the same drug [13,15].

In our study, the sequence Moxe before vemurafenib plus rituximab was used. However, it is not clear today, which regimen should be used first in HCL patients relapsed after PNAs. Currently, Moxe is approved by the FDA and EMA for the treatment of relapsed HCL patients and is commercially available in the USA, whereas it is not clear whether and when it will be commercialized elsewhere. Vemurafenib or dabrafenib are not approved for this indication yet, and are used off label also in the USA, but are marketed for *BRAF*-mutated melanoma worldwide and the efficacy of vemurafenib plus rituximab seems higher [17] than that of Moxe [11]. For these reasons, in patients with early relapsed or refractory HCL outside of clinical trials we recommend Moxe or vemurafenib plus rituximab depending on local availability and ease of reimbursement of the respective drug(s). On the other hand, vemurafenib seems to be an effective and well-tolerated drug in neutropenic patients with infection [26,27,29,30]. In such patients vemurafenib or dabrafenib, if available, might be used before Moxe, as there is lesser experience with the latter drug in the setting of an active infection. However, Moxe does not directly worsen neutropenia and in fact neutropenia is usually disease-related and resolves during cycle 2. Future studies should clarify the optimal treatment sequence with these drugs in HCL patients.

## 5. Conclusions

In the coming years, targeted drugs will play a key role in standard therapy for patients with relapsed or refractory HCL after treatment with PNAs. The history of the presented cases indicates that novel targeted agents and, in particular, vemurafenib plus rituximab, improve the prognosis of HCL patients, even those who are heavily pretreated with PNAs and Moxe. However, despite the interesting perspective of targeted treatments in HCL, older options should not be forgotten, including bendamustine plus rituximab [31] and (pegylated) interferon [20]. Still, HCL remains an incurable disease in such an advanced relapsed/refractory disease setting, and the development of novel therapeutic strategies is required.

## Figures and Tables

**Table 1 jcm-10-02800-t001:** Patient characteristics and treatment results.

	Age at Diagnosis/Sex	Therapies Before Moxe *	Moxe Treatment *	Vem + R Treatment	Response to V + R *	Subsequent Treatment *
1	28/F	1. 2-CdA (PR; 21 m)	40 µg/kg iv × 3 every 28 days × 6 (PR; 22 m)	Vem 240 mg BID for 16 weeks + R 375 mg/m^2^/d/iv/every 2 weeks × 8	MRD negative CR (ongoing at 38 m)	None
		2. 2-CdA + R (PR; 6 m)				
		3. IFN-α (PR; 17 m)				
		4. 2-CdA (PR; 26 m)				
2	33/F	1. 2-CdA × 2 courses (PR; 15 m)	40 µg/kg iv × 3 every 28 days × 6 (PR; 11 m)	Vem 960 mg BID for 2 weeks, (intolerance), dose reduction to 240 mg BID for 14 weeks + R 375 mg/m^2^/d/iv/every 2 weeks × 8	Hematologic response (18 m)	Vem 240 mg BID for 16 weeks + R 375 mg/m^2^/d/iv/every 2 weeks × 8 (MRD positive CR; 17 m). Dabrafenib (75–150 mg/d BID po) + trametinib 2 mg/day indefinitely (ongoing hematologic remission; 2 m)
		2. IFN-α (NR)				
		3. 2-CdA + R (PR; 6 m)				
3	53/M	1. 2-CdA (CR; 18 m)	40 µg/kg iv × 3 every 28 d × 6 (CR MRD-; 17 m)	Vem 240 mg BID for 16 weeks + R 375 mg/m^2^/d/iv/every 2 weeks × 8	MRD negative CR (13 m)	Vem 240 mg BID for 7 weeks + R 375 mg/m2/d/iv/4 doses; Splenectomy (PR; 5 m)
		2. 2-CdA (CR; 27 m)				Vem 240 mg BID for 16 weeks + R 375 mg/m2/d/iv/every two weeks × 8 (hematologic remission; 4 m)
		3. 2-CdA (CR; 98 m)				Dabrafenib (75–150 mg/d BID po) + trametinib 2 mg/day indefinitely (hematologic remission; 10 m)
4	45/M	1. 2-CdA (PR; 22 m)	40 µg/kg IV × 3 every 28 d × 6 (CR MRD+; 38 m)	Vem 960 mg BID for three weeks, Only 1 dose of R due to infection	Not evaluable	Died after three weeks of treatment with vemurafenib due to pneumonia and septic shock
		2. 2-CdA (PR; 27 m)				
		3. 2-CdA (CR; 144 m)				
		4. 2-CdA (PR; 23 m)				
		5. 2-CdA (PR; 16 m)				
		6. 2-CdA + R (PR; 24 m)				

Abbreviations: BID—two times a day; 2-CdA—2-chlorodeoxyadenosine, cladribine; CR—Complete response; F—Female; IFN-α—Interferon alpha; PR—Partial response; m—Months; M—Male; Moxe—moxetumomab pasudotox; MRD—Minimal residual disease NR—No response; R—Rituximab; Vem—Vemurafenib. * duration of hematologic response was measured in months from the end of treatment.

## Data Availability

The data presented in this study are available from the corresponding author for request.

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
