# Peer review of "Vemurafenib and Rituximab in Patients with Hairy Cell Leukemia Previously Treated with Moxetumomab Pasudotox"

_jcm, 2021, doi:10.3390/jcm10132800_

Round 1
Reviewer 1 Report
1- In Page three, authors stated; Before Moxe initiation, all patients had been previously treated with at least three "regimens" and one patient with six lines of therapy. I think the word regimens should be changed to lines as some patient were treated with the same regimen multiple times rather than different regimens.
2- In Page four, QOD x3 is better spelled out.
3- Case descriptions could be shortened since all treatments were described in the table.
4- In page seven, recommend removing the following statement from the second paragraph in the discussion; and the Moxe-rituximab combination may prove to be better than the single agent.
Author Response
Reviewer 1.
Comments and Suggestions for Authors
- In Page three, authors stated; Before Moxe initiation, all patients had been previously treated with at least three "regimens" and one patient with six lines of therapy. I think the word regimens should be changed to lines as some patient were treated with the same regimen multiple times rather than different regimens.
Response: The word regimens has been changed to lines
- In Page four, QOD x3 is better spelled out.
Response: Spelled out as “Moxe at 40 micrograms on Days 1, 3, and 5 of each 28‑day cycle for up to five-six courses within a phase 3 clinical trial”.
- Case descriptions could be shortened since all treatments were described in the table.
Response: Case descriptions have been shortened as requested, especially in the parts concerning earlier 2-CdA based therapies.
- In page seven, recommend removing the following statement from the second paragraph in the discussion; and the Moxe-rituximab combination may prove to be better than the single agent.
Response: Removed as requested

Reviewer 2 Report
The authors describe four relapsed/refractory HCL patients who were treated by vemurafenib plus rituximab after receiving moxetumomab pasudotox. The article is well written and very useful for the medical community.
For improving the manuscript, please:
1) Shorten the length of the article. it is not useful to be so exhaustive and the long description is redundant with table1. Table 1 should be improved in quality.
2) Patient 4 died three weeks after starting vemurafenib and rituximab from severe bacterial infection. The patient received high dose (960 mgtwice daily) of vemurafenib, as published by Tiacci E et al. Do you recommend high or low dose of vemurafenib?. The other 3 patients received low dose. What is the explanation?.
3) Combining vemurafenib plus rituximab is effective. Two patients achieved CR and negative MRD. Unfortunatly the patients relapsed very quickly. What do you recommend for testing MRD?
4) Vemurafenib plus rituximab was introduced very late and after Moxe in all cases. Do you recommend the sequence Moxe then vemurafenib plus rituximab or the reverse? Discuss and tell us your opinion on the optimal drug sequencing.
5) Two patients were re-treated with the same combination and the results are encouraging, particularly in case 3. Do you recommend a strategy of stop and go, as previously published?.
Author Response
Reviewer 2.
The authors describe four relapsed/refractory HCL patients who were treated by vemurafenib plus rituximab after receiving moxetumomab pasudotox. The article is well written and very useful for the medical community.
For improving the manuscript, please:
- Shorten the length of the article. it is not useful to be so exhaustive and the long description is redundant with table1. Table 1 should be improved in quality.
Response: The article has been shortened as requested, mainly by reducing the case description. The quality of the table 1 is improved.
- Patient 4 died three weeks after starting vemurafenib and rituximab from severe bacterial infection. The patient received high dose (960 mgtwice daily) of vemurafenib, as published by Tiacci E et al. Do you recommend high or low dose of vemurafenib?. The other 3 patients received low dose. What is the explanation?
Response: The first three cases were treated with lower doses of vemurafenib, as we assumed that they are safer in heavily pretreated patients and equally effective as higher doses as suggested in the paper by Dietrich at al (Dietrich et al. BRAF inhibition in hairy cell leukemia with low-dose vemurafenib. Blood. 2016, 127, 2847-2855). However, we increased the doses in the fourth patients when the excellent preliminary results of the higher doses of vemurafenib in combination with rituximab in Tiacci et al study became available. Furthermore, Patient 4 died early of a severe infection that was clearly related to HCL and not to vemurafenib.
- Combining vemurafenib plus rituximab is effective. Two patients achieved CR and negative MRD. Unfortunatly the patients relapsed very quickly. What do you recommend for testing MRD?
Response: Actually only one of the two patients relapsed quickly (patient 3, after 13 months; patient 1 instead is relapse-free after 38 months; patient 2 was not analyzed for MRD after therapy). Every technique for MRD testing has pros and cons and we would recommend to test them all, i.e. flow cytometry and allele-specific (ideally digital) PCR for the BRAF-V600E mutation in the bone marrow aspirate, as well as immunohistochemistry for BRAF-V600E in the bone marrow biopsy. In patient 3, because conventional allele-specific PCR was used (with an analytical sensitivity of 0.1% mutant alleles), it might be that MRD was actually present below 0.1% (and that digital PCR could have identified it). We have accordingly added this statement in the Results (page 5) and the Discussion (page 7).
- Vemurafenib plus rituximab was introduced very late and after Moxe in all cases. Do you recommend the sequence Moxe then vemurafenib plus rituximab or the reverse? Discuss and tell us your opinion on the optimal drug sequencing.
Response: We have added the following text in the discussion: “In our study, the sequence Moxe before vemurafenib plus rituximab was used. However, it is not clear today, which regimen should be used first in HCL patients relapsed after PNAs. Currently, Moxe is approved by FDA and EMA for the treatment of relapsed HCL patients and is commercially available in the USA, whereas it is not clear whether and when it will be commercialized elsewhere. Vemurafenib or dabrafenib are not approved for this indication yet, and are used off label also in the USA, but are marketed for BRAF-mutated melanoma worlwide and the efficacy of vemurafenib plus rituximab seems higher [17] than that of Moxe [11]. For these reasons, in patients with early relapsed or refractory HCL outside of clinical trials we recommend Moxe or vemurafenib plus rituximab depending on local availability and ease of reimbursement of the respective drug(s).On the other hand, vemurafenib seems to be effective and well tolerated drug in neutropenic patients with infection [26,27,29,30] In such patients vemurafenib or dabrafenib, if available, might be used before Moxe, as there is lesser experience with the latter drug in the setting of an active infection. Future studies should clarify the optimal treatment sequence with these drugs in HCL patients„ .
- Two patients were re-treated with the same combination and the results are encouraging, particularly in case 3. Do you recommend a strategy of stop and go, as previously published?
Response:
Yes, we think that it is a reasonable option

Reviewer 3 Report
Interesting case series about 4 patients with classical HCL. Case 1,2 and 4 had often suboptimal responses and are therefore the more "resistant" type of cases seen fortunately in a minorty of cases. The novelty of the paper relates to responses after moxtumumab pasudotox which is a helpful information in these cases. The paper is well written.
- Some inconsistencies exist and need to be revised (eg. case 1 i.v. and iv. first line treatment: 5 days? Sept 2012 iv or sc....).
- The table is very hard to read and needs to be edited. I recommend to do it in a timeline manner (starting at diagnosis, each treatment/BM written in 90° from the time axis (eg.: 2CdA ; 2CdA+R, Ven960...; treatment above and BM below or in the axis (eg 60%HCL). The exact duration could be added at the bottom of the table.
- sIL2R is an easy available parameter in HCL cases and correspondes very well with HCL-mass. If these values are availble in these cases, it would give an helpfull additional information.
- treatment in HCL can be divided in limited (2CdA, Moxetumumab., Rituximab...) and unlimited (IFNa). The duration of vemurafnib treatment is not fully clear in HCL. A concept like induction with high doses followed by low dose maintenance could be of help especially in cases like the ones presented with not many option left. Please elaborate the pros and cons.
- conclusions: despite the interesting perspective of targeted treatments in HCL older options should not be forgotten for patients with many relapses (eg. bendamustin). In addition pegylated Interferon would be an interesting drug for maintenance after achieving CR.
Author Response
Reviewer 3.
Interesting case series about 4 patients with classical HCL. Case 1,2 and 4 had often suboptimal responses and are therefore the more "resistant" type of cases seen fortunately in a minorty of cases. The novelty of the paper relates to responses after moxtumumab pasudotox which is a helpful information in these cases. The paper is well written.
- Some inconsistencies exist and need to be revised (eg. case 1 i.v. and iv. first line treatment: 5 days? Sept 2012 iv or sc....).
Response: We carefully revised the paper and corrected inconsistencies
- The table is very hard to read and needs to be edited. I recommend to do it in a timeline manner (starting at diagnosis, each treatment/BM written in 90° from the time axis (eg.: 2CdA ; 2CdA+R, Ven960...; treatment above and BM below or in the axis (eg 60%HCL). The exact duration could be added at the bottom of the table.
Response: The table has been re-edited and it seems to be mor clear than earlier version.
- sIL2R is an easy available parameter in HCL cases and correspondes very well with HCL-mass. If these values are availble in these cases, it would give an helpfull additional information.
Response: Unfortunately, these data are not available
- treatment in HCL can be divided in limited (2CdA, Moxetumumab., Rituximab...) and unlimited (IFNa). The duration of vemurafnib treatment is not fully clear in HCL. A concept like induction with high doses followed by low dose maintenance could be of help especially in cases like the ones presented with not many option left. Please elaborate the pros and cons.
Response: We have added the paragraph on this topic in the discussion part (“We used vemurafenib for limited time as in the most previous studies of this drug [13,14,17]. Limited treatment duration can decrese adverse events noted with BRAF inhibitors including skeletal pain, photosensitivity, skin tumors and renal toxicity. On the other hand treatment with vemurafenib, especially in combination with rituximab induces deep, long responses making unlimited treatment redundant [17]. In addition, relapsed patients can achieve response after retreatment with the same drug [13,15].”
- conclusions: despite the interesting perspective of targeted treatments in HCL older options should not be forgotten for patients with many relapses (eg. bendamustin). In addition pegylated Interferon would be an interesting drug for maintenance after achieving CR.
Response: Thank you for this interesting point. We have added bendamustine and interferon as options not to be forgotten in the Conclusions.

Reviewer 4 Report
In this report authors described experience with BRAF-inhibitor vemurafenib in combination with rituximab in patients with relapsed heavily treated hairy cell leukemia. Experience with this combination is limited, several small phase 2 and retrospective studies were published, and description of use in a real life practice is valuable. Additional interest is in prior use of other experimental drug moxetumomab after repeated cycles of established treatment with cladribine and interferon with or without rituximab. Vemurafenib was given in lower than in other studies doses. Retreatment with this regimen was feasible. This data may help in establishment management, schedule and sequence of treatment lines in HCL patients failed to standard agents.
The text is written clear, seems that case description has redundant text (see also table) and could be cleaned up a bit.
Author Response
Reviewer 4
In this report authors described experience with BRAF-inhibitor vemurafenib in combination with rituximab in patients with relapsed heavily treated hairy cell leukemia. Experience with this combination is limited, several small phase 2 and retrospective studies were published, and description of use in a real life practice is valuable. Additional interest is in prior use of other experimental drug moxetumomab after repeated cycles of established treatment with cladribine and interferon with or without rituximab. Vemurafenib was given in lower than in other studies doses. Retreatment with this regimen was feasible. This data may help in establishment management, schedule and sequence of treatment lines in HCL patients failed to standard agents.
The text is written clear, seems that case description has redundant text (see also table) and could be cleaned up a bit.
Response We thank the Reviewer for positive review of our paper. We have shortened the case reports as requested, especially in the 2-CdA treatment parts..

Round 2
Reviewer 2 Report
The revised form of the manuscript was improved and now the manuscript is useful for each of us and easy to read.
In the present form it is acceptable for publication.